# Pension and Active Ageing: Lessons Learned from Civil Servants in Indonesia

**Aris Ananta [1,2,\*], Ahmad Irsan A. Moeis [3], Hendro Try Widianto [3], Heri Yulianto [3] and Evi Nurvidya Arifin [2]**

1  Faculty of Economics and Business, Universitas Indonesia, Ul Depok 16424, Indonesia
2  Centre for Advanced Research, Universiti Brunei Darussalam, Bandar Seri Begawan BE1410, Brunei;
   evi.arifin@ubd.edu.bn
3  Directorate General of Budget, Ministry of Finance, Jakarta 10710, Indonesia;
   irsan.moeis@kemenkeu.go.id (A.I.A.M.); hendro.widianto@kemenkeu.go.id (H.T.W.);
   heri.yulianto@kemenkeu.go.id (H.Y.)
*  Correspondence: arisananta@ui.ac.id

**Abstract:** Many developing countries are currently facing an ageing population without sufficient preparation for old-age financial adequacy, an important component in active ageing. One question is whether a pension system can create old-age financial adequacy. At the same time, many countries are shifting their pension systems from a defined benefit to a defined contribution pension system to improve the welfare of older people while maintaining state budget sustainability. Indonesia is not an exception. This paper learns from civil servants in Indonesia, where the retirement payout from the existing pay-as-you-go, defined benefit system is meagre. The system is to be transformed into a defined contribution one. Using a simulation method, this paper examines whether the proposed system will provide a better retirement payout, which is higher than the minimum wage and will allow retirees to maintain their pre-retirement income. This paper concludes that the proposed system alone is not sufficient to create old-age financial adequacy and, therefore, is less able to contribute to active ageing. To improve the retirement payout, among other things, the retirement age should be raised and made optional, and the accumulated savings should be re-invested during the retirement period.

**Keywords:** active ageing; financial adequacy; pension system; civil servants; standard of living

## 1. Introduction

Many developing countries in Asia are currently facing an ageing population without sufficient preparation for active ageing. "Active ageing" is a framework used by WHO (2002), viewing old age as the opportunity to be healthy, to participate in society, and to be secure (including economic security). This framework has been widely followed in Europe since 2005 as a policy response to challenges brought about by the ageing population (Foster and Walker 2014; Principi et al. 2018). This concept refuted the "dependency" concept, which emphasizes the passivity of older people, as elaborated by Foster and Walker (2021). Instead, active ageing stresses the importance of activity and that older people are not simply recipients or even burdens to society.

The active ageing concept has triggered the debate on delaying retirement age, extending employment, and working towards the sustainability of the pension system (Foster 2018). Earlier, Foster (2012) argued that pension policies should be able to provide options for older people that would allow them to work and create old-age financial adequacy, which is a crucial element in active ageing. However, ILO and ASEAN (2020) reported that no country in Southeast Asia has reached the same stage of old-age financial adequacy as those in Europe. In Southeast Asia, where Indonesia is located, the retirement payout is not sufficient to let older people live actively, as defined in the active ageing framework. At most, the pension functions as a subsidy or an "accessory" to the old-age financial

adequacy. Older people still need financial support from their own employment and/or transfers from families, communities, or other government interventions.

In contrast, a generous pension system was originally seen as a celebration of the success of modern western countries to reward their older people. This system was put in place to avoid poverty in old age. The generous retirement payout is financed by the current taxpayers—a system called pay-as-you go (PAYG). However, this system becomes questionable on the ground of state financial sustainability, when the ratio of the number of pension recipients to number of taxpayers is becoming higher and rising rapidly, as a consequence of an ageing population (Danzer et al. 2016; Mertl et al. 2019; Wang 2021).

Therefore, these countries have begun to search for an alternative pension system and how to restructure their pension system to provide adequate financial support for older people, but also guarantee state budget sustainability. Along with the emergence of neo-liberal economic policies, an emphasis on individual responsibility following market mechanism, there is an increasing trend among European or OECD countries to shift from a defined benefit, PAYG pension system, to a defined contribution pension system. Unlike in the PAYG system, where the responsibility of providing adequate pension is in the hands of government, the defined contribution pension system puts the responsibility of creating old-age financial adequacy in individuals themselves. The employees contribute to the savings, sometimes topped up by the employers, which are then to be invested by the employers and to be distributed back to the employees when they retire or during paid retirement period. (OECD 2016; Foster 2018).

Nevertheless, shifting to a defined contribution system, where individuals actually save for themselves, may not necessarily create old-age financial adequacy. Lin et al. (2021) indicated that the shift to the defined contribution system is often problematic. They suggested a reform within the PAYG by reducing the pension benefit, starting from the new cohort. Their study showed that this reform within PAYG brought about higher economic growth and welfare in the long run.

With large informal employment, an underdeveloped pension system, and an antici-pated ageing population, Indonesia faces similar challenges in its pension system. Civil servants account for the majority of public services remuneration in many countries, in-cluding Indonesia. They are the only large group covered by a formal and stable pension system. Yet, the number of civil servants has been declining mostly because of rapidly increasing number of new retirees (Asian Development Bank 2021). To improve the welfare of the retired civil servants through its pension system, the government of Indonesia has been considering restructuring the pay-as-you-go (PAYG), defined benefit system used for civil servants, the military, and the police.

Under the existing pension system for civil servants, the amount of monthly retirement payout is meagre. The government regulation no. 18 in 2019 on the basic pension of civil servants and their spouses[1] stated that the highest monthly retirement payout of a director at grade IV/d, the second highest rank, is only IDR 4,246,300 or US 303.3. This amount is slightly lower than the minimum wage in the capital of Jakarta (IDR 4,276,344), which is the highest provincial minimum wage in Indonesia. Therefore, to help create state budget sustainability in anticipation of an ageing population and to create old-age financial adequacy, the restructuring plan proposes to shift the system into a defined contribution system. With this proposed reform, the responsibility of providing a pension is shifted from the government to the individuals themselves, guaranteeing state budget sustainability.

This paper examines to what extent the proposed defined contribution system for civil servants can contribute to active ageing through old-age financial adequacy (as an indicator of economic security) in Indonesia, a developing Asian country anticipating an ageing population.

This paper addresses three questions regarding old-age financial adequacy in the proposed defined contribution system. First is whether the retirees will receive better payout than in the existing pay-as-you-go system. Second is whether the retirees will receive decent payout, measured by a higher payout than the capital city's minimum wage,

the highest provincial minimum wage in Indonesia. Third is whether the retirees will be able to maintain their pre-retirement standard of living.

This paper conducts a literature review and a set of simulations to examine whether the proposed defined contribution pension system can help civil servants to age actively through the expected retirement payout. The simulation uses a present value approach, which considers inflation rate to calculate purchasing power of the future payout. This simulation also assumes that the accumulated savings are re-invested during the retirement period, to prevent declining purchasing power of the payout as the retirees age.

The findings are expected to provide lessons for other groups in Indonesia or other developing countries, who want to transform its pension system into defined contribution systems. The assumptions in other groups may be different, but the important lesson from this study is that a defined contribution system alone is not necessarily sufficient to create old-age financial adequacy and therefore is less likely to let the retirees age actively.

The next section is a literature review, discussing old-age financial adequacy, a defined contribution pension system, active ageing, and Indonesian pension system. The third section deliberates the method and assumptions of the simulation. The fourth section provides the results of simulation. The paper is closed with concluding remarks, discussing the main conclusion from the simulation and policy recommendations derived from the active ageing framework, beyond pension system.

## 2. Literature Review

### 2.1. Old-Age Financial Adequacy

Old-age financial adequacy can be a household decision, as older people may live with other people in the same household (Gomes et al. 2021). Furthermore, it is not easy to determine old-age financial adequacy. What is "adequate"? People's needs can be unlimited. One often used measurement of old-age financial adequacy is income replacement ratio, indicating a percentage of old-age financial support to pre-retirement final or average whole-life income. Countries such as China, India, Pakistan, and Singapore use a progressivity approach to allow those with low pre-retirement income to enjoy a higher income replacement ratio. The ratio is especially low among high income earners in Singapore because there is a ceiling on the contribution to accumulated savings to be invested for retirement payout (Gee and Fong 2019).

The retirement income ratio varies. Brady (2010) argued that the ratio should be lower than the pre-retirement income (less than 1) considering three assumptions as follows. First, retirees usually do not spend much for transportation, clothes, and food, compared to when they were still working. However, the assumption that retirees do not spend much on transportation cost is questionable. Transportation cost is still needed to continue actively participating in society in old age. The rising old age tourism is another important issue related to leisure in later life. Second, retirees do not buy many durable goods. Again, this assumption can be examined further, as the durable goods may depreciate and need to be replaced. Third, retirees no longer need to finance their children. This assumption may be correct, but some older people may continue to transfer money to their children and even grand-children. Finally, the major weakness of Brady's calculation is that it does not include rising expenditure on healthcare. It is therefore possible that the needed retirement income ratio can be even larger than 1. Indeed, Brady also argued that the retirement income ratio among the poor in the US may be very high, approaching 1 or higher, because they have small amounts of pre-retirement income.

As defined by ILO and ASEAN (2020), old-age financial adequacy is the amount of financial support providing a decent life in old-age, regardless of the pre-retirement income. Old-age financial adequacy should not depend on pre-retirement performance in the labour market. The need to de-couple the labour market performance from old-age financial adequacy is also suggested by Rappaport and Bajtelsmit (2019). They argued that people who did not obtain relatively large amounts of pre-retirement income saved much

less, and therefore their retirement payout may be very small—they may fall into poverty, unless they receive transfers from family, community or the state.

Rappaport and Bajtelsmit (2019) proposed old-age financial adequacy with three criteria: the abilities to maintain the pre-retirement standard of living, to cover future rising expenses (especially due to health expenditure and inflation rate), and to live at least above the poverty rate to allow a decent standard of living.

### 2.2. Defined Contribution Pension System

The main goals of the pension policy are to create state budget financial sustainability and provide old-age financial adequacy. Most older people's financial support in European countries rely on retirement payout which is mostly financed by the current taxpayers. This is the so-called pay-as-you-go (PAYG) system. However, the rapidly ageing population in Europe has threatened the state budget sustainability as the percentage of labour force (tax payers) to older people (recipients of pension) becomes smaller. European countries have therefore faced double challenges in its pension system: to maintain state budget sustainability and to create old-age financial adequacy. Old-age financial adequacy is measured with the ability to prevent poverty, whether the pension payout can replace pre-retirement income, and the number years the older people receive the pension payout (European Commission 2017).

The defined benefit pension system, which is accompanied by the PAYG system in OECD public pension arrangement, has also been challenged by their ageing population, low economic and wage growth rates, and low rate of return of investment. As a result, in addition to the reform on the PAYG system itself, the use of a defined contribution pension system has been widely observed to solve state budget sustainability (OECD 2016). Furthermore, the COVID-19 pandemic has aggravated the challenges, as also seen in many countries in the world. The health crisis has resulted in people unable to save sufficiently for old-age. Many countries have then expanded their job retention schemes and unemployment benefits, and allowed early withdrawal of savings guided by regulation to ensure the sustainability of retirement savings arrangement (OECD 2020).

On the other hand, in the defined contribution pension system, the old-age financial adequacy depends much on the performance of employees in the labour market. As shown by Anderson and Klinger (2016), the retirement payout depends on how much they contribute when they are working, the rate of return on the invested saving, and amount of administrative cost to invest the saving. If the workers have high income and save a lot, they will have a large amount of accumulated saving at the beginning of their retirement periods, and therefore receive higher retirement payout throughout their retirement periods. People who do not work full time or have fragmented employment, especially among women, may end up having a small amount of accumulated savings at the start of and throughout their retirement periods. They may then be less likely to age actively.

Finland is an example of a European country with a defined benefit pension system that provides generous welfare for their older people while maintaining fiscal sustainability with minimal cost. Yet, the main challenge for the pension system is also its ageing population. One of the solutions is to raise retirement age—to shorten the span of time to provide the payout and lengthen the span of time that the older people can still contribute to the economy and tax revenue (Valkonen 2020). Similarly, Latulippe and Turner (2019) recommended raising retirement age for United States and Canada.

Traditionally, older people in Asia relied on family and community for old-age financial adequacy. However, demographic changes and modernization may have resulted in declining family and community supports and rising need for formal pension systems (Leng 2019). In contrast to rich European countries, Asian emerging countries, such as China, India, Thailand, Vietnam, and Indonesia, face their ageing population before they become rich. Not surprisingly, many older people in Southeast Asia are still working to

maintain their standard of living, as the existing pension system is weak and not generous (Arifin and Ananta 2009; Teerawichitchainan et al. 2019).

Similar to the European countries, Asian countries with a PAYG system, such as South Korea and China, may also face challenges in their state budget sustainability. They may reduce the retirement payout and/or raise retirement ages. On the other hand, there has been a gradual tendency in the direction of universality in social protection, financed by taxpayers, even in "productivist" countries such as Hong Kong, Taiwan, South Korea, and Singapore (Gee and Fong 2019).

As the concern on state budget sustainability rose because of ageing population, the concept of active ageing provides an alternative to view ageing population from a positive side, rather than as dependency or burden for the society. In the active ageing framework, a pension should be de-coupled from the labour market—retirement payout should not be related to individual labour market performance as in a defined contribution pension system. A pension system should also reward unpaid jobs. One alternative is to create a universal, unconditional, adequate old-age financial adequacy, financed by taxpayers. This alternative may require raising the productivity (perhaps by introducing more technology) of the taxpayers (Foster 2018).

*2.3. Active Ageing*

The term "active ageing" is adopted by WHO (2002) to recognize that long life should be seen as opportunities to enhance the quality of life of older people through improving health, participation in the society, and security. Health is here meant in a broad sense, including increasing healthy years of life, health promotion and healthy life styles since early years of life, and social support for lonely older people. Participation includes social and economic activities with the family and communities, long-life education and training, age-friendly transportation and information. Security includes protection, safety and dignity of older people; and old-age financial adequacy, a covering social safety net for those who are poor and alone, as well as personal planning for old age since young ages.

The concept of active ageing refutes the traditional concept of old-age dependency, where older people are seen as a burden for the society when they reach 60 or 65 years old and above. An ageing population is viewed as a threat to the economy. On the other hand, the active ageing concept sees older people as an asset for the society, not a burden; an ageing population is a celebration, not a threat (WHO 2002; Cylus et al. 2019).

As emphasized in WHO (2015), health is the most important factor in realizing the potential benefit from longevity. Having longer years of life filled with better health, older people have more freedom to be and to do what they want. In contrast, if longer years of life are accompanied by poor health, the older people and the society as a whole will be deprived from opportunities to enjoy the longevity—they, and the society, may suffer from the ageing population.

Active ageing emphasized a life-course approach. Policies to improve wellbeing of older people should start at a very early stage of life, since the time of the prospective parents. As discussed by Aegon (2017), health is closely related to old-age financial adequacy, and therefore having healthy lifestyle throughout the life course will contribute to active ageing, such as for travelling, having more time with families or friends, developing hobbies and part-time working. On the other hand, financial adequacy may improve health, measured in longer expectancy of life (Walczak et al. 2021). Furthermore, Cheng and Chan (2018) concluded that continuing to work may slow-down the deterioration of health of older people.

Finland has also followed and implemented active ageing policies. Active older people are visible everywhere. This reduces the cost of "maintaining" the older people. Even, the older people can still contribute directly to the economy (thisisFinland n.d.). Nordmyr et al. (2020) showed that older Finish did not see themselves as passive or dependent. They had a positive mindset. They remained active, showing an active ageing society.

Many Asian countries followed and adopted the active ageing framework. However, the level of implementation varied, in particular among Southeast and East Asian countries as classified by Walker and Aspalter (2015). Although it was still preliminary, they classified these countries into four groups. First is a group of countries where the governments support a narrow definition of active ageing and rely more on community, family, and individuals themselves, rather than through active/pro-active government social intervention. Examples are Singapore and Hong Kong. Second is a group of countries where the governments follow a broader concept of active ageing, but do not have the political will for active/pro-active government social intervention. Taiwan is an example. Third is a group consisting of countries which follow a broad concept of active ageing. They have political will for active/pro-active government social intervention. However, results are still waiting to be seen. Examples are Korea, China, Malaysia, and Indonesia. The fourth group are countries with a broad concept of active ageing, and have been successful in their implementation of the concept. They also have active/pro-active government social interventions, being inclusive and following life-course approach. However, there have been no countries in this "ideal" fourth group.

Therefore, Cylus et al. (2019) argued that ageing population is not necessarily harmful for the economy and society. With active ageing policies, caring for older people is not costly, as they can take care of themselves. They can still contribute to the economy and society. They may have unpaid work or the work cannot be monetized. They may also contribute more as they are no longer burdened to take care of their children. With active ageing, older people create the so-called "silver economy". In short, active ageing itself has economic benefits. People can work longer and better. More older people can participate in the labour force and social, non-paid activities.

Foster and Walker (2021) concluded that policies on active ageing should pay attention to at least eight aspects. First, policies should be able to contribute to the overall well-being of older people. Unpaid jobs such as volunteering and caring should be seen as important as paid jobs. Second, policies should emphasize promotive health policies, rather than curative health policies, on the basis of a life-course approach. This means promoting healthy older people starts from an early life, even at an embryonic stage. Third, policies should be inclusive. For example, it is not restricted to young-older people to produce employment, but including the old-older people, who are frail. Fourth, policies should consider inter-generational solidarity, to avoid generational conflicts of interest. For example, should retirement age be raised, to increase older people's opportunity to earn more money while delaying the promotion of the younger people? Is there any win–win solution? Fifth, policies should be made on the rights and obligations of older people in relation to issues such as social protection and long-life learning. Sixth, policies should be able to empower the older people, making older people resources for the society, rather than a burden on the society. Seventh, policies should respect diversity, and not oppress the marginalized and disadvantaged. Eight, policies should be adaptable and flexible as people characteristics are dynamic, changing with time, conditions, and locations.

### 2.4. Pension System in Indonesia

As described by OECD (2019), Indonesia has been successful in creating a social protection system, replacing the traditional poverty alleviation approach. It uses social assistance programmes and reforms social insurance system (including expansion of coverage and augmentation of the benefit). The implementation of universal health coverage (BPJS Kesehatan) is an example in which citizens contribute premiums, but the Government subsidizes fully or partly in order that everybody can receive the same benefit.

Moeis et al. (2019) showed that from a budget perspective, Indonesia has been moving closer to a social-democrat model, at least until 2014, the latest data they had. In the social-democrat model, the role of state is much larger than those of market and family, similar to Scandinavian countries, in contrast to conservative and liberal models. Furthermore, they believed that the trend continues during the Joko Widodo presidency's first term

(2014–2019) and current term (2019–2024). Therefore, the pension system is only part of a much larger social protection system. Discussions on the pension system should be contextualized at the overall social protection system.

Nevertheless, Ananta et al. (2021) found out that older people in Indonesia may have difficulty in having financial adequacy, to maintain the pre-retirement standard of living. The younger generation (future older people) wanted to work longer as they perceived that the accumulated saving would not be adequate to finance their old-age. In other words, being forced to retire when a person is still able and willing to work is not a celebration. Consequently, older people mostly rely on private transfer, especially from spouses and children.

The Indonesia pension system is to be well-developed. Only about 14.0% of people aged 60 years and over is covered by the system in 2015. Furthermore, the covered people are mostly from the high income quantile, including those who work as civil servants, military, and police (ILO and ASEAN 2020).

Indonesia has a separate pension system for its civil servants, military, and policy. It is mandatory for civil servants, military, and police to participate in a defined benefit pension system. The employees pay monthly premium for retirement. State-owned PT Taspen manages the pension for civil servants; and state-owned PT Asabri for military and police. It has two schemes. First is *Akumulasi Iuran Pension* (AIP—Accumulated Premium Pension). The premium is set at 4.75% of the basic salary. Although the premium is managed by PT Taspen and PT Asabri, the retirement payout is paid from the state-budget.

Second is *Tabungan Hari Tua* (THT—Old Age Saving). The premium is 3.25% of the basic salary. The amount of benefit received by the retirees does not depend on the contribution of the employees. It is paid at the start of retirement period, or when they become disabled during the working period. It is paid to the beneficiaries when they die, before or during the retirement period. It should be noted that in these two schemes, the accumulated saving is not re-invested during the retirement period.

Pension system for private sector consists of both mandatory and voluntary participation. The mandatory participation is managed by state-owned BPJS *Ketenagakerjaan* (Labour). It has two old age pension schemes. First is *Jaminan Pensiun* (JP—Pension Protection), a defined contribution system, paid monthly during the retirement period or lump sum payment at age 56, the beginning of retirement period, for formal sector employment only. Second is *Jaminan Hari Tua* (JHT—Old Age Benefit), a defined contribution system, with monthly payment during retirement or a lump sum payment at the beginning of retirement period, or the employee dies/stops working before retirement. It covers both formal and informal sector employment (BPJS Ketenagakerjaan n.d.).

The voluntary participation is managed by *Dana Pensiun Pemberi Kerja* (DPPK—Employer's Pension Fund) and *Dana Pensiun Lembaga Keuangan* (DPLK—Financial Institution Pension Fund). DPPK is managed by the firms for their employees. It is a non-mandatory defined contribution system, with investment risk born by the employees. DPLK is not related to employment. It is conducted by private financial institutions such as banks and insurance companies.

## 3. Simulation Method and Assumptions

### 3.1. Method

This paper uses simulations of employees' life throughout the working and retirement periods. All retirement payouts are calculated in their present values. This calculation evaluates the future purchasing power in 2020 prices, as the value of money in the future is less than the value of money today. This is to break the "money illusion", as if having a larger amount of money, yet the purchasing power is much lower. The present value of the payout in the simulation scenarios can then be compared with the existing, pre-retirement level of basic salary and take-home income.

During the working period, employees contribute monthly premiums or pension contributions to be invested by state-owned enterprises. This results in an accumulated

saving by the end of the working period. The investment is carried out yearly using a compound technique. An innovation in our simulation is that the calculation of the premium is based on take-home income, rather than the basic salary as in the current defined benefit, PAYG system. In the existing system, take home income consists of basic salary, standard allowance, and other allowances. Other allowances widely vary. The basic salary in the existing system is much below the take-home income, resulting in very low premium to be saved and invested and therefore low payout.

This accumulated saving is then distributed during a 20-year paid retirement period, and there will not be any payout after this period.[2] Another innovation is made by allowing re-investment of the accumulated saving. Without re-investment, the nominal payout will be the same from year to year, resulting in a declining present value of the payout as the retirees age, erased by inflation. This innovation anticipates rising expenditure because of declining health as the retirees become older.

The third innovation is that when the retirees die before the end of the 20-year retirement period, the spouses/relatives will continue receiving the full payout. In the existing system, the spouses/relatives do not receive the full payout.

At the start of the retirement period, the accumulated saving is divided by 20 years × 12 months to obtain the monthly payout during the first year of retirement. The remaining saving is then re-invested, with the same rate of return as that in the working period. At the start of the second year of retirement, the new accumulated saving is divided by 19 years × 12 months, to have monthly payout during the second year of retirement. The remaining payout is again re-invested. The same procedure is carried out until the last year of the 20-year paid retirement period, except that the remaining accumulated saving is completely spent in the last paid retirement year.

The simulated payout is then compared with the existing payout to examine whether the proposed system is better or worse than the existing one. Second, it is also compared with the capital city (Jakarta) minimum wage, the highest among provinces in Indonesia, to examine whether or not the retirees are out of poverty in the proposed system. The highest minimum wage is selected as the threshold of poverty because it is not applied to the whole Indonesian population, but to the few who are working as civil servants. As reported in *Kompas* (2019), working as civil servants is a dream for many people, as it offers a steady job and income, health insurance, and pension. The civil servants are joining the rising middle class in Indonesia. Third, the payout is also compared with the pre-retirement take-home income to learn whether the retirees can maintain the pre-retirement standard of living.

The simulation takes two scenarios of civil servants. One is from the best career path, where the civil servant ends up being a Director before retiring. Another one is a middle career path, where the civil servant finishes as a Head of Section before retiring. The two paths are selected as they may represent the best civil servants. This paper examines whether people with these paths can live well in the proposed defined contribution system. Others may face more challenging times during their retirement.

All payouts are presented in both Indonesian rupiah (IDR) and US dollars. An alternative is to present the purchasing power parity (PPP) of the US dollar, because USD 1.00 in Jakarta, Indonesia, may buy more goods and service than in New York. As described by IMF (in Callen n.d.), this argument is especially true when the consumption does not involve internationally traded goods.

However, Indonesians, including and particularly the civil servants, have been increasingly consuming internationally traded goods and services. Indonesia, with its 270 million population, has become a lucrative market for many international goods and services (Australian Government n.d.). Furthermore, as mentioned earlier, civil servants are part of the important middle class in Indonesia. Therefore, the civil servants may not be limited to consume locally non-traded goods, but they consume many internationally traded goods and services as Indonesia is open to international trade. As a result, this paper does not present the payout in the purchasing power parity (PPP). Nevertheless, as a reference for those who want to examine the payout in PPP, US$1 is equivalent to IDR 4674 in 2020.[3]

*3.2. Assumptions*

For convenience, the present value is calculated at the base year of 2020 prices. Employees are assumed to start working at age 25 in 2020 with an undergraduate degree and receive salary level of III/a based on the government regulation (PP) no. 15/2019 for basic salary; and no 156/2014 for allowances. The simulation uses two scenarios for the career path. The first path is for employees who later obtain master degrees and end up as Directors, receiving salary grade IV/d, the second highest rank in civil service. They will have 35 years of working period when they retire at age 60 in 2055. The second path is for employees who remain holding an undergraduate degree, works for 33 years, end with grade III/d as Heads of Section, and retire at age 58 in 2053.[4]

In each career path scenario, the accumulated premium during employment is invested yearly. The path of basic salary follows the government regulation no. 15/2019; and standard allowance follows government regulation no. 156/2014. The simulation assumes three scenarios for "other allowances": 10%, 15%, and 20% from the sum of both the basic salary and standard allowance. The premium paid has three scenarios. First is the low premium scenario, with 10% of take-home income (and with other allowances at 10% of the basic salary and standard allowance). Second is the medium premium scenario with 15% of take-home income (and with other allowances at 15% of the sum of the basic salary and standard allowance). Third is the high premium scenario with 20% of take-home income (and with other allowances at 20% of the sum of the basic salary and standard allowance).

As mentioned earlier, payout is only paid during the first 20 years of retirement, not providing anything for those who can live longer. If the retirees die before the age of 80 or 78, the spouse or relatives will receive a full payout in the remaining 20-year period. There will be no more payout when the retirees reach 80 or 78 years old. As people live longer, future studies should relax this assumption, although it may lower the monthly payout.

The simulation also has three scenarios for investment rate of return,[5] both in the working and retirement periods: low rate (6% annually), medium rate (9% annually), and high rate (12% annually). During 2013–2020, as reported by PT Taspen (n.d.), the rate of return has been varying from 8.0% to 9.0%. Finally, the simulated future payout is discounted to the value in 2020, using a constant discount rate of 4.0% annually, as the average inflation rate in Indonesia during 2010–2020 was 4.32%.[6] For example, a monthly payout of IDR 20.28 million or USD 1448 in 2055 is equivalent to only IDR 5.0 million or USD 357 in 2020, assuming USD 1 is constant at IDR 14,000 for 35 years.

The simulation assumes constant aspiration throughout the working and retirement periods, as there is no data on changing aspiration. Higher aspirations may make the retirees worse financially and much less likely to have active ageing. Further studies should take this issue into consideration.

## 4. Results for Career Ending as a Director

*4.1. Existing System*

Based on the existing system, Table 1 clearly shows that the basic salary of a Director holding a master degree with a salary level IV/d at the end of working period is very low, only IDR 5.5 million or USD 392 monthly. However, the take-home income is much higher, about five times the basic salary.

**Table 1.** Take-home income and monthly retirement payout of director, IV/d existing system, 2020 (*in thousand IDR*).

| | Take-Home Income | | | | Minimum Monthly Payout | Maximum Monthly Payout |
|---|---|---|---|---|---|---|
| | Basic Salary | Standard Allowance | Other Allowance | Total | | |
| IDR | 5489 | 21,330 | 2682 | 29,501 | 1561 | 4246 |
| USD | 392 | 1524 | 192 | 2107 | 111 | 303 |

Notes: Jakarta minimum monthly wage is IDR 4276 thousands in 2020. Source: Based on government regulations no. 15/2019 on basic salary; and no. 156/2014 on standard allowance. Other Allowances are assumed to be 10% of the sum of both basic salary and standard allowance. The payout is regulated through government regulation no. 18/2019.

Under the existing pension system, the Director will retire with a meagre amount of payout, at maximum of IDR 4.25 million or USD 303 monthly, which is even smaller than the basic salary and slightly lower than the Jakarta minimum wage (IDR 4.28 million), the highest among provinces in Indonesia. This is a drastic decline from IDR 29.50 million, the last pre-retirement take-home income. The director will not live with the pre-retirement standard of living and even below the minimum wage of the capital city, Jakarta, if the retired director only relies on the payout from the civil servant system.

The loss of standard of living, even falling below minimum wage, during retirement brings about far reaching social and health consequences. The retiree is much less likely to age actively due to the income constraint. This may partly explain why Indonesian older people mainly do home bound life activities such as watching television, reading and less travelling (Arifin et al. 2012; Badan Pusat Statistik 2020).

*4.2. Low Premium Scenario*

The director who starts to retire at the age of 60 under the scenario of low premium and the highest rate of investment will receive the present value of the monthly payout at IDR 5.94 million in the first year of retirement (Table 2). This is just slightly higher than the basic salary received in the final year of working period (Table 1). However, this simulated payout is better than the payout provided under the existing pension system (IDR 4.25 million). The simulated payout is also higher than Jakarta minimum wage (IDR 4.28 million) in 2020. As the minimum wage varies across provinces and the value of payout is the same wherever the retirees reside, some may consider migrating to another province (Rachmawati and Chotib 2018) such as Yogyakarta, which has the lowest provincial minimum wage (IDR 1.7 million), to have a better purchasing power during retirement. Yogyakarta is perceived to be one of the best places for retirement[7].

**Table 2.** Present values of monthly retirement payout, Director, IV/d (*in thousand IDR).*

| Investment Rate of Return | Premium | | |
|---|---|---|---|
| | Low | Medium | High |
| **6%** | | | |
| First Payout (60 years old) | IDR 1841 | IDR 2883 | IDR 4001 |
| | $132 | $206 | $287 |
| Last Payout (79 years old) | IDR 2644 | IDR 4141 | IDR 5761 |
| | $189 | $296 | $412 |
| **9%** | | | |
| First Payout (60 years old) | IDR 3209 | IDR 5028 | IDR 6995 |
| | $229 | $359 | $500 |
| Last Payout (79 years old) | IDR 7801 | IDR 12,270 | IDR 17,072 |
| | $559 | $876 | $1219 |
| **12%** | | | |
| First Payout (60 years old) | IDR 5940 | IDR 9311 | IDR 12,955 |
| | $424 | $665 | $925 |
| Last Payout (79 years old) | IDR 24,283 | IDR 38,064 | IDR 52,958 |
| | $1734 | $2719 | $3783 |

Note: rupiah is expressed in thousands. Low premium: 10% × take-home income. Medium premium: 15% × take-home income; High premium: 20% × take-home income. Source: calculated by the authors.

Under this scenario, although living above the minimum wage, the retired director will not be able to maintain the pre-retirement standard of living, as the simulated payout is much lower than the pre-retirement take-home income. This may result in financial insecurity. As many older people are not in good health (Arifin 2015), and therefore facing higher health expenditure, the retired director may suffer worse financial insecurity. Moreover, if the accumulated savings are not reinvested, the retiree will become more miserable over time, especially when there are no other financial sources. The retired director will receive the same nominal payout during the 20-years retirement period. Therefore, the present value of the payouts will decline as the retired director ages.

By re-investing the remaining accumulated saving, the simulation shows that the present value of the monthly payout will rise, when the retirees become older. For instance, at the age of 79, with the highest rate of return (12%), the present value of monthly payout can be IDR 24.28 million, or 82.31% of pre-retirement take-home income. It can be relatively large by the end of the paid retirement period. Yet, at age 79, the retirees may spend more for their health. If the retirees can be active through their old age, this amount of money may give an assurance for financial security. In other words, if the retired Director follows scenarios of lower rate of return (either 6.0% or 9.0%), the daily life activities can be affected severely.

### 4.3. Medium Premium Scenario

The medium premium scenario together with the highest rate of return provides the payout in the first year of retirement of IDR 9.31 million per month, which is more than double the existing payout and much higher than the basic salary. Nevertheless, it is only about one-third of the pre-retirement take-home income, which can still prevent the retirees from having better health and active life. Without re-investment, the payout will become smaller as the retired director ages, and make the retiree much less likely to age actively.

With re-investment, the monthly payout at the end of the 20th year of retirement will be IDR 38.06 million, larger than the final pre-retirement take home income. In other words, by the end of the paid retirement period, approaching 79 years old, the retired Director may enjoy a better standard of living than prior to the retirement, assuming the health condition remains the same throughout the retirement period. With a lower rate of return (9.0% and below), the retired Director will suffer from a lower level of living standard in the whole retirement period, making this retiree less able to age actively.

### 4.4. High Premium Scenario

The combination of large "other allowances", high premium and high rate of return provides the highest payout in the first year of retirement than other combinations/scenarios. The retired Director will receive the monthly payout of IDR 13.00 million, which is three times the existing payout, or more than double the basic salary. Yet, it is still far below (43.91%) the last pre-retirement take-home income. In this best scenario (high premium scenario), the retiree will still be far below the pre-retirement standard of living in the first year of retirement. Without re-investment, the retired director will face more difficulty in maintaining the pre-retirement standard of living.

As time passes toward older age and the accumulated savings continue to be invested, the retiree will enjoy a much higher standard of living. Based on this scenario, at the age of 79, the payout can be IDR 52.96 million, even almost double the pre-retirement standard of living. Assuming the health expenditure will not rise more than double as the retiree ages, the retired Director will be more likely to age actively.

## 5. Results for Career Ending as Head of Section

### 5.1. Existing System

The government regulations no. 15/2019 stated that the basic salary of the Head of Section by the end of the working period is only IDR 4.80 million. This is about 12 percent higher than the Jakarta minimum wage. However, total allowance is much higher (Table 3). As a result, the take-home income is about three times the basic salary.

**Table 3.** Take-home income and monthly retirement payout of Head of Section, III/d based on the existing system, 2020 *(in thousand IDR).*

|  | Basic Salary | Standard Allowance | Other Allowance | Take-Home Income | Minimum Monthly Payout | Maximum Monthly Payout |
|---|---|---|---|---|---|---|
| IDR | 4797 | 8458 | 1326 | 14,581 | 1561 | 3598 |
| USD | 343 | 604 | 95 | 1041 | 111 | 257 |

Note: Jakarta minimum monthly wage is IDR 4276 thousands. Source: It is based on the government regulations no. 15/2019 on the basic salary; and no. 156/2014 on standard allowance. Other Allowances are assumed to be 10% of the sum of both basic salary and standard allowance.

When the Head of Section retires at the age of 58, under the existing system, with the government regulation no. 18/2019 on premium pension, this person will only receive a maximum monthly payout at IDR 3.60 million, which is about 75 percent lower than the basic salary and 84 percent lower than minimum wage in Jakarta. Furthermore, the payout is of course much lower than (precisely 25% of) the pre-retirement take-home income. That means the existing pension system will force the retired Head of a Section to lower the standard of living drastically. The retiree may even live in poverty as the payout is below Jakarta minimum wage. As time passes, the payout declines in its present value. This retiree is much less likely to age actively if he/she does not have any other financial sources.

### 5.2. Scenarios of Future Payout

The same pattern is seen between the Head of Section and the Director that the monthly payout in the first year of retirement is lower than the pre-retirement take-home pay for whatever the scenarios of simulation. The retired Head of Section will receive a higher monthly payout in the last year of retirement (the 20th year of retirement, at age 77), only when the remaining accumulated saving is re-invested. Without re-investment, the present value of the monthly payout will decline as the retiree ages. The difference between the payout in the first year and 20th year of retirement becomes larger with a higher investment rate of return and a higher premium (Table 4).

In the best scenario, a rate of return at 12% and a high premium, the present value of the last payout reached IDR 41.14 million, more than four times the present value of the first payout at IDR 10.1 million. In this scenario, the payout in the first year is already much better than the payout in the existing system, and is closer to the last pre-retirement take-home income. The payout is above the minimum wage of Jakarta, and the retiree may even be able to maintain the standard of living, as it is closer to the last pre-retirement take-home income. The retiree's wellbeing will possibly be better at older ages, and ultimately will be likely to age actively, assuming that the health status does not deteriorate.

**Table 4.** Present value of monthly retirement payout Head of Section, III/d *(in thousand IDR)*.

| Investment Rate of Return | Premium | | |
|---|---|---|---|
| | **Low** | **Medium** | **High** |
| **6%** | | | |
| First Payout (58 years old) | IDR 1398 $100 | IDR 2192 $157 | IDR 3050 $218 |
| Last Payout (77 years old) | IDR 2007 $143 | IDR 3148 $225 | IDR 4379 $313 |
| **9%** | | | |
| First Payout (58 years old) | IDR 2487 $178 | IDR 3901 $279 | IDR 5427 $388 |
| Last Payout (77 years old) | IDR 6070 $434 | IDR 9520 $680 | IDR 13,245 $946 |
| **12%** | | | |
| First Payout (58 years old) | IDR 4613 $329 | IDR 7233 $517 | IDR 10,064 $719 |
| Last Payout (77 years old) | IDR 18,856 $1347 | IDR 29,570 $2112 | IDR 41,141 $2939 |

Note: Low premium: 10% × take-home income; Medium premium: 15% × take-home income; and High premium: 20% × take-home income. Source: calculated by the authors.

However, if both the investment rate of return and premium are low, the present value of the payout is much smaller (IDR 1.40 million in the first year of retirement and IDR 2.00 million in the last year of retirement) than the payout given by the existing system. This clearly shows misery of life during retirement. This is an undesirable scenario.

The middle scenario seen in Table 4 is with the rate of return at 9.0% (the currently observed rate of investment) and medium premium. In the first year of retirement, the retired Head of Section will receive IDR 3.90 million monthly, which is lower than the basic salary and lower than the Jakarta minimum wage; though higher than the payout in the existing system. Moreover, the payout is much below the last pre-retirement take-home income. The retired Head of Section will be tough to maintain the standard of living in the first year of retirement. Even with the high premium, the first payout can only be IDR 5.43 million, less than half of the last pre-retirement income take-home income.

With the re-investment of the accumulated saving, the payout will be better (IDR 9.52 million) based on the medium premium scenario at the end of paid retirement period. However, this payout is 65% lower than the last pre-retirement take-home income. Even without health deterioration, the retiree under this career path is less likely to be able to maintain a pre-retirement standard of living, though at age 77, the difference between the payout and last pre-retirement take-home income is getting smaller.

## 6. Concluding Remark

This paper provides lessons from the expected pension reform in Indonesia's civil servants, particularly with respect to its contribution on active ageing through financial adequacy. The most important lesson is that the reform from a defined benefit pension system to a defined contribution system is not necessarily sufficient to let retirees age actively through their old-age financial adequacy.

Under the existing defined benefit pension system for civil servants, the military, and the police in Indonesia, the monthly payout for the retirees is meagre, even below the capital city minimum wage. However, Indonesia is preparing to shift to a defined contribution system. The question is whether the proposed change can help the retirees age actively. Specifically, the questions are whether the proposed defined contribution system can provide better retirement payout, whether it can make the retirees live out of poverty (higher than Jakarta minimum wage, the highest provincial minimum wage in Indonesia), and whether it allows the retirees to enjoy the pre-retirement living standard.

The paper concludes that the payout in the proposed system can provide a better payout than that in the existing, pay-as-you-go system, except in scenarios with low investment rate of return (6.0%), and medium investment rate of return (9.0%) combined with low premium. It can help the retirees out of poverty only with investment rate of return at least 9.0% combined with medium premium for the retired Director, and high premium for the retired Head of Section. Civil servants with lower levels than the Head of Section may face more challenging time to maintain pre-retirement standard of living, and even to live with pension payout above the capital city's minimum wage.

Moreover, regardless of the investment rate of return and premium scenarios, the payout in the first year of retirement remains much below the amount needed to maintain the pre-retirement standard of living. Even for a retired Director with the highest investment rate of return (12.0%), the retiree will only have higher standard of living (compared to the pre-retirement) by the end of paid retirement period. This better standard of living assumes that the retiree's health does not deteriorate during the first 20 years after retirement. However, this assumption may not be tenable for many older people.

Therefore, this paper concludes that reliance on the pension system alone is not sufficient to help the retirees having old-age financial adequacy. In turn, the retirees are less likely to age actively—to be healthy, able to participate in society's socio-economic activities, and to be socially as well as economically secured. There should be other ways to improve the wellbeing of the civil servant retirees, including changing the mindset into the one that older people are social and economic assets, rather than a burden or a dependency. There should not be any barrier for older people to participate in social and economic activities, as participation is one of the rights of older people.

This paper provides three recommendations to enable civil servants to age actively. First is through increasing the monthly retirement payout. Second is through raising transfer from families/friends, societies, and government. Third, is going beyond the pension system and embracing the broader active ageing framework. This active ageing framework, with its life-course approach, will particularly benefit the younger generations—who are the future older people.

### 6.1. Improving the Retirement Payout

First, the accumulated saving should be re-invested during the retirement period. Otherwise, the saving will decline in value overtime.

Second, the premium should be based on take-home income, not basic salary. The basic salary is too small compared to the take-home income.

Third, the investment rate of return during working and retirement period should be targeted at 12.0%, higher than the current rate of return (9.0%).

Fourth, retirement age should be raised and made optional, allowing employees to accumulate more savings. Meritocracy should also be considered, neither seniority nor juniority, along with the higher retirement age.

Fifth, the national inflation rate should be stabilized at a low rate, to raise the present value of the payout.

Sixth, take-home salary should be increased, along with continuous growth of the economy.

Seventh, by raising labour productivity, taxpayers can be much more productive, and the government can have more money for direct and indirect social assistances to the older people.

### 6.2. Enhancing Transfer from Families, Communities and Government

First, policies can be made to revive the disappearing extended family networking. Digital technology can connect physically far-away relatives, creating a new "super extended family" network.[8] This network can function to help each other, sharing information and relieving the financial burden of this family members.

Second, policies can be designed to encourage philanthropic activities, including religious ones such as *zakat* and *wakaf* in Islam. A generous tax-deduction can be offered to those providing contribution/philanthropic activities, including non-monetary contribution.

Third, a universal assistance to older people can be considered to guarantee that older people live above minimum wage. Special assistance on healthcare for older people should also be considered.

*6.3. Beyond Pension: Active Ageing Framework*

An active ageing framework can help to create old-age financial adequacy. When retirees are healthy, able to participate in society, and secured, the need of government assistance will be less. The retirees themselves are better in contributing to their own financial adequacy, by having (more) income and/or less health expenditure. Below are three recommended policies for consideration.

First, policies should be made following a life-course approach to promote healthy life styles since an early life. Economic policies should be conducive to the creation of healthy life styles.

Second, policies should create older people-friendly infrastructure in the public place, housing complex, and within the house to minimize older people's mobility capability deprivation.

Third, policies should be made to equip older people with digital skill and produce older people friendly digital technology. With the ability to utilize digital technology, older people will become empowered and improve their quality of life.

**Author Contributions:** Conceptualization: A.A.; methodology: A.A., H.T.W. and H.Y.; formal analysis: A.A., A.I.A.M. and E.N.A.; data curation: A.A., H.T.W. and H.Y.; writing—original draft preparation: A.A., A.I.A.M., H.T.W. and H.Y.; writing—review and editing: A.A. and E.N.A.; project administrator: A.A. All authors have read and agreed to the published version of the manuscript.

**Funding:** This project received no external funding.

**Institutional Review Board Statement:** Not applicable.

**Informed Consent Statement:** Not applicable.

**Data Availability Statement:** The data are already mentioned under "source" in each table.

**Conflicts of Interest:** The authors declare no conflict of interest.

## Notes

[1] The government regulation no. 18/2019 on "Penetapan Pensiun Pokok Pegawai Negeri dan Janda/Dudanya" can be accessed from this link https://peraturan.go.id/common/dokumen/ln/2019/pp18-2019.pdf (accessed on 10 October 2021).

[2] Longer period of paid retirement will mean lower payout. Further studies should examine longer paid retirement period as Indonesians are living longer.

[3] https://data.oecd.org/conversion/purchasing-power-parities-ppp.htm (accessed on 11 October 2021).

[4] Retirement age for civil servants is 58 years old, meaning that they start the retirement period at age 58. However, higher level civil servants such as a Director can work until 59 years old, and start retiring at age 60.

[5] Investment rate of return is also often called "yield on investment (YOI)".

[6] https://www.bi.go.id/id/statistik/indikator/data-inflasi.aspx (accessed on 13 October 2021).

[7] https://jogja.tribunnews.com/2018/02/10/kota-kota-yang-nyaman-ditinggali-saat-pensiun (accessed on 20 October 2021).

[8] See Ananta (2020) for a discussion on the rising digital super-extended family system.

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
