# Peer review of "Pension and Active Ageing: Lessons Learned from Civil Servants in Indonesia"

_socsci, doi:10.3390/socsci10110436_

Round 1
Reviewer 1 Report
This is an extremely interesting article, with three noteworthy pension scenarios.
It is worth remembering that the article is addressed to an international audience and, as such, needs to reflect certain conditions and interdependencies in Indonesia. It is worth locating on the map both the highest and lowest rates of poverty in Indonesia, as well as the location of the discussed places in relation to the capital.
It is also worth discussing pension systems in Europe, especially in Finland, which is considered to have a highly efficient one. Please consider introducing an analysis of European pension systems and their efficiency in the literature review. It is necessary to review the literature, point out the advantages of the different systems, and the possibility of their implementation in Indonesia.
It is also worth noting that aspirations are rising in changing societies, especially in the group shown. This results in growing expectations not only in relation to the quality of food or places to live, but also leisure activities, holidays, and increasing one's material base (eg. sustainable resources). These should be taken into account.
It would be useful to compare the identified groups with others, or at least to signal whether they would benefit in a similar way as the study group. The author has chosen a specific group, but would the benefits be similar for other ones?
Please indicate how the grades are understood (line 64).
In order for the international reader to understand the relationships in Indonesia, it is also worth showing what can be purchased in the country at a particular pension level. Indicating the value in dollars is not sufficient because the purchasing power of money is different in each country. It is worth presenting this in the text.
Likewise useful would be indicating whether the transition to more favourable systems for pensioners can be shouldered by the pension system in Indonesia. Where will the funds for this come from?
Please explain what is meant by active ageing.
Above all else, the section discussing the literature on the subject is worth expanding as there are basically no references to the literature.
Author Response
Deaar Editors,
Attached in the response to Reviewer 1
Thank you.
Best regards,

Reviewer 2 Report
In MDPI journals you can find publications that can be used in the reviewed article. For example in Risks (Risks 2021, 9(4), 65; https://doi.org/10.3390/risks9040065).
Author Response
Dear editors,
Attached is our response to Reviewer 2
Thank you.
Best regards,

Reviewer 3 Report
The paper evaluates expected retirement income adequacy from the Indonesia Pension's system.
Main Comments:
The list of references is lacking and nearly non-existent. There is plenty of work on this topic globally and none of it is cited. Gomes et al (2021) is a place to start. What research has been done on Indonesia Pensions? The reader is left to find out for themselves how the paper places in greater context on pension research.
As the analysis and three research questions the paper focuses on is on retirement income adequacy, the abstract, introduction and framing of the paper should be around this topics instead of active ageing - which is secondary to the main topic. The authors move towards this in pages 5 and 7, but it should be at the beginning.
Retirement income should be discussed in relative terms. For example, what does Rp. 4,246,300 or US$303.3 mean in context of retirement income for these individuals? Does this represent 80% or 110% of pre-retirement income? It is difficulty to understand the state of the Pension system without discussing the relative measures.
Tables 1-3 should at least have a measure of relative payouts. i.e. min/max should be shown as a percent of pre-retirement salary and absolute monetary terms.
The final sections of the paper needs substantial revisions. Some comments are not substantiated by the paper:
- "Fifth, the national inflation rate should be kept low, to raise the present value". This is beyond the scope of the paper.
- "Better technology". How is this related to the main analysis of the paper?
The review advises that the paper be thoroughly proof-read prior to submission, or to go through a copy-editor.
Minor:
Page 5: "Indonesia has been successful in creating a social protection system" -- this is conjecture with no citations to support such a claim.
traditional poverty alleviation approach
Page 5: X showed that.
Reference List: X References.
Gomes, F., Haliassos, M., & Ramadorai, T. (2021). Household finance. Journal of Economic Literature.
Author Response
Dear editors,
Attached is our response to Reviewer 3.
Thank you.
Best regards,

Round 2
Reviewer 3 Report
Much improved.